# *APOE* ε4-Allele in Middle-Aged and Older Autistic Adults: Associations with Verbal Learning and Memory

**DOI:** 10.3390/ijms242115988

**Published:** 2023-11-05

**Authors:** Samantha A. Harker, Lamees Al-Hassan, Matthew J. Huentelman, B. Blair Braden, Candace R. Lewis

**Affiliations:** 1School of Life Sciences and Psychology, Arizona State University, Tempe, AZ 85287, USA; saharker@asu.edu; 2College of Health Solutions, Arizona State University, Tempe, AZ 85287, USA; lalhassa@asu.edu (L.A.-H.); bbbraden@asu.edu (B.B.B.); 3Neurogenomics Division, Translational Genomics Research Institute, Phoenix, AZ 85004, USA; mhuentelman@tgen.org

**Keywords:** autism, aging, genomics, cognition, learning, memory, *APOE*, Alzheimer’s disease, genetics, neurobiology

## Abstract

Autism spectrum disorder (ASD) is a neurodevelopmental disability and recent evidence suggests that autistic adults are more likely to develop Alzheimer’s disease (Alz) and other dementias compared to neurotypical (NT) adults. The ε4-allele of the Apolipoprotein E (*APOE*) gene is the strongest genetic risk factor for Alz and negatively impacts cognition in middle-aged and older (MA+) adults. This study aimed to determine the impact of the *APOE* ε4-allele on verbal learning and memory in MA+ autistic adults (ages 40–71 years) compared to matched NT adults. Using the Auditory Verbal Learning Test (AVLT), we found that ε4 carriers performed worse on short-term memory and verbal learning across diagnosis groups, but there was no interaction with diagnosis. In exploratory analyses within sex and diagnosis groups, only autistic men carrying *APOE* ε4 showed worse verbal learning (*p* = 0.02), compared to autistic men who were not carriers. Finally, the *APOE* ε4-allele did not significantly affect long-term memory in this sample. These findings replicate previous work indicating that the *APOE* ε4-allele negatively impacts short-term memory and verbal learning in MA+ adults and presents new preliminary findings that MA+ autistic men may be vulnerable to the effects of *APOE* ε4 on verbal learning. Future work with a larger sample is needed to determine if autistic women may also be vulnerable.

## 1. Introduction

By 2030, there will be approximately 700,000 elderly autistic adults with a formal diagnosis in the U.S. [1]. Autism Spectrum Disorder (ASD) is a neurodevelopmental disability identified by social communication challenges as well as restrictive and repetitive behaviors and interests [2,3]. Recently, the Centers for Disease Control and Prevention (CDC) estimate the prevalence of autism diagnoses in children aged eight years old to be 1 in 36 in the United States, with the prevalence in boys approximately three times higher than in girls [4]. Importantly, autistic individuals experience more health-related vulnerabilities and premature mortality compared to neurotypical (NT) adults. Findings from healthcare records show that middle-aged and older (MA+) autistic adults are at a higher risk of developing Alzheimer’s disease (Alz) and related dementias when compared to non-autistic individuals [5,6]. Additionally, previous studies suggest that autistic individuals are more likely to develop cognitive problems as they age [6,7,8]. Better understanding MA+ autistic adults’ aging vulnerabilities and their relation to Alz is vital for providing the best care for autistic adults across the lifespan.

Alz is a progressive neurodegenerative disorder associated with cell death and ultimately reduces cognitive abilities and causes dementia [9,10]. ASD and Alz share similar symptoms, such as cognitive and communicative impairment, insomnia, weak muscular interaction, and speech and hearing challenges [11,12]. In a series of two publications, our research group recently showed preliminary longitudinal findings that MA+ autistic adults demonstrate accelerated short-term memory, long-term memory, and hippocampal volume loss, compared to matched NT adults [13,14]. Taken together, MA+ autistic adults may have increased vulnerability towards accelerated cognitive decline and increased risk for developing Alz compared to NT adults. 

The *APOE* gene provides instructions for making a protein called apolipoprotein E, a lipid transport protein involved in neuronal repair and cholesterol transport. The various *APOE* alleles are differentiated by two collocated single nucleotide polymorphisms in *APOE*’s coding regions [15,16]. The ε2-allele shows evidence of protection against Alz, the ε3-allele is considered the most common allele [17], and the ε4-allele is considered the strongest genetic risk factor for sporadic Alz yet discovered [17,18,19]. Interestingly, others have found that autistic individuals are more likely to carry the ε4-allele [20], although this has not been shown when assessing entire families with an autistic individual versus families without an autistic individual [21].

Even before dementia presents, ε4-allele carriers have worse cognitive performance compared to non-carriers and some studies show sex differences. For example, healthy older adults who carry the ε4-allele perform more poorly than non-carriers on verbal learning and memory tests [22,23]. Carriers of the ε4-allele may have a higher risk for ASD-like symptoms in childhood [24] as well as greater risk for cognitive decline [23]. Interestingly, ε4-allele carriers may experience memory decline ten years earlier than non-carriers, at 60 years old and 70 years old, respectively [25]. Further, male ε4-allele carriers, exclusively, present with greater beta-amyloid plaque burden, worsened verbal memory ability, decreased hippocampal volume, and brain hypometabolism [26]. Notably, when cognitive decline begins, women can retain verbal memory for longer periods than men [25,27,28]. Past case-control studies have indicated that the ε4-allele and its correlation to Alz may be more prevalent in women, in addition to other neurodegenerative brain changes such as widespread brain hypometabolism and cortical thinning [29,30]. Understanding sex differences in the impact of ε4-allele status on cognitive aging may contribute to early precision interventions for the ASD community.

The present study examined the effect of *APOE* allele status on verbal learning and memory in MA+ autistic adults, compared to matched NT controls. We hypothesized that MA+ autistic adults who are *APOE* ε4-allele carriers will have worse verbal learning and memory abilities compared to ASD ε4-allele non-carriers and NT controls, regardless of allele status. Finally, as an exploratory analysis, we evaluated if sex moderates the *APOE* ε4-allele carrier status effect on verbal learning and memory in autistic and NT adults.

## 2. Results

There was a main effect of *APOE* ε4 for short-term memory and verbal learning, with ε4 carriers performing worse across diagnosis groups (Table 1 and Table 2, Figure 1 and Figure 2). The *APOE* ε4-allele did not significantly affect the participants’ long-term memory performance. The interaction between autism diagnosis and ε4-allele carrier status was not significant for any verbal learning and memory measure. For verbal learning, sex was a significant predictor (Table 2); therefore, exploratory analyses separating diagnosis groups by sex were conducted. For autistic males, NT males, and NT females, carriers and non-carriers were compared via *t*-test. Only autistic males carrying *APOE* ε4 showed worse verbal learning compared to autistic male non-carriers (Table 2, Figure 3). Due to the small sample size of autistic female non-carriers (*n* = 2), single-case analyses were conducted, and there were no differences between each non-carrier and the group of carriers (Table 2). See Appendix A. for all group means and standard deviations.

## 3. Discussion

This study is the first to investigate the *APOE* ε4-allele’s effect on cognition in MA+ autistic adults compared to matched NT adults, specifically investigating verbal learning and memory. We replicated previous literature indicating that the *APOE* ε4-allele has a significant negative impact on cognition. In exploratory analyses, we compared the impact of the *APOE* ε4-allele in autistic male and NT males and females on verbal learning, since previous studies suggest that sex/gender influence ASD, Alz, and the effect of *APOE,* respectfully [31,32,33,34]. We put forward new findings showing that only autistic male *APOE* ε4 carriers had a worse performance in verbal learning abilities, while this was not the case for NT males or females. In separate single-case Bayesian analyses, our two female autistic non-carriers also did not show significant differences from female autistic carriers. 

Our results replicated known associations indicating that ε4-allele carriers perform worse on verbal learning tasks. For example, a study by Liu et al. [33] reported that middle-aged *APOE* ε4-allele carriers performed worse on verbal learning tasks compared to NT controls. However, for short-term memory, there was less evidence that *APOE* ε4 has a negative impact, with one study reporting benefits in short-term memory performance during midlife exclusively for male ε4 carriers [35]. Alternatively, we reported worse short-term memory performance of ε4 carriers; this discrepancy may be explained by the wider and older age range of the participants in this study. Further, our cohort was comprised of both autistic and NT adults, which may have impacted our findings since we previously reported that MA+ autistic adults are more likely to show clinically meaningful decline in short-term verbal memory compared to NT controls [13]. 

Lastly, other studies reported a negative impact of the ε4-allele on long-term verbal memory performance [36], while we found no ε4-allele effect on this measure. In some cases, such as Caselli et al., 2015 [37], the discrepancy may be because of age differences, as our sample was younger and past research has shown the effects of the *APOE* ε4-allele to be sex- and age-dependent [38,39]. Additionally, our previous research has shown that autistic adults are not vulnerable to accelerated long-term verbal memory decline, as they are with short-term verbal memory [13]. Future research with larger sample sizes is needed to determine if these discordant short-term and long-term verbal memory findings are being driven by autistic adults.

We reported novel findings that within sex and diagnosis groups, the ε4-allele negatively impacts verbal learning performance in autistic male adults, but not in NT males or females. Due to only two autistic female non-carriers, this sex by diagnosis group was compared through single-case analyses and neither was found to be different from the group of autistic female carriers. These results should be interpreted with caution, and future research is warranted to determine if sex and ASD diagnosis may moderate the impact of the ε4-allele on verbal learning. Past case-control studies have indicated that the ε4-allele and its correlation to Alz may be more prevalent in women [40], with higher co-incidence of the two [38,40], and that autistic females have higher self-reported rates of cognitive decline in dementia screenings than autistic men [8]. However, when evaluating the effects of *APOE* ε4 on cognitive function between men and women, others have shown men to be more vulnerable to ε4 effects than women [39], including effects on hippocampal volume and hypometabolism in the mildly and cognitively impaired brain [25]. Our verbal learning findings extend this to show that autistic males may be especially vulnerable to *APOE* ε4 effects on cognition. This may be related to general sex differences in verbal learning and memory, where both autistic and NT female adults tend to perform better than autistic and NT male adults [41]. Past research suggests that performance in non-social cognitive areas is sex-dependent in autistic adults [42]. Further, ASD females may perform better in verbal tasks and demonstrate faster processing speeds than their ASD male counterparts [41,42,43]. Therefore, it is critical to further evaluate the detrimental effects of the *APOE* ε4-allele on cognition in autistic males and females as they are more likely to develop age-related cognitive problems [13,20] and early-onset Alz [6].

### Limitations

This study investigated *APOE’s* association with cognition in ASD, with several limitations worth noting. First, our sample included only autistic adults with average to above average IQs and therefore does not represent the full spectrum of cognitive abilities in autistic individuals. Second, the small sample size may be underpowered. Our sample only had two autistic female ε4-allele non-carriers, which necessitated single-case Bayesian analyses, which are less reliable than group comparisons. Future research should include more autistic females to evaluate the three-way interaction between ASD diagnosis, *APOE* allele status, and sex. Additionally, future research with greater statistical power should employ multivariate analyses to investigate the role of demographic factors (e.g., participant health history, race/ethnicity, education level, mental and physical activity levels, and familial health history) on these results. Lastly, a larger sample size could evaluate the effect of ε4 dose (i.e., homozygotes vs. heterozygotes), presence of an ε2, or each possible *APOE* allelic combination on learning and memory in autistic adults, which was not possible in this study. 

## 4. Methods and Materials

### 4.1. Participants

Study demographics are summarized in Table 3. Appendix A summarizes additional participant health demographics. Sex was defined as assigned at birth, which was concordant with all participants’ gender identity in this sample. Participants were recruited between the years 2014 and 2022 and were partially representative of participants from previous publications [5,13,44,45]. Recruitment strategies included flyers posted around Arizona, USA in a 30-mile radius, community partners, the Southwest Autism Research & Resource Center (SARRC) Phoenix, Arizona, USA database, and word of mouth. The SARRC database is voluntary and includes information about individuals who have been involved in previous clinical or research projects at SARRC. Participants in both groups underwent the same screening and enrollment procedures.

### 4.2. Inclusion/Exclusion Criteria

Autistic participants had their diagnosis formally verified at SARRC with the Autism Diagnostic Observation Schedule-2, module 4 (ADOS-2; [46]) and a brief psychiatric history interview administered by a research-reliable psychometrist. A score ≥ 7 on the ADOS-2 and an assessment by a psychologist with 25 years of ASD diagnostic experience confirmed DSM-5 criteria were met for their ASD diagnosis. NT participants were excluded if they had a first-degree autistic relative, were suspected or confirmed to have an ASD diagnosis, or if they had a T-score > 66 on the Social Responsiveness Scale-2 Adult Self-Report (SRS-2; [47]). Participants from both groups were excluded if their full-scale IQ score was <70 on the Kaufman Brief Intelligence Test-2 (KBIT-2) [48], they scored <25 on the Mini Mental State Exam (MMSE; [49]), or they self-reported a neurological disease such as a stroke or dementia, a head injury with loss of consciousness, known genetic disorders, a substance use disorder, or current use of seizure medications. Comorbid psychiatric conditions were non-exclusionary because of their high prevalence in the ASD population [50,51,52,53].

### 4.3. Verbal Learning and Memory 

Participants performed the Rey Auditory Verbal Learning Test (AVLT; [49]). The AVLT consists of a supra-span word list of 15 words which are repeated five times (A1–A5), followed by a free recall trial after a 20–30-min delay (A7). Raw scores for short-term immediate recall (A1; short-term memory), and long-term delayed recall (A7; long-term memory), as well as total words (A1–A5; learning) were used for analyses.

### 4.4. APOE Genotype

Participants provided saliva samples (Oragene|OG-600) during standard lab visits. DNA was extracted using the Oragene’s *DNA* purification protocol and reagents. DNA underwent polymerase chain reaction (PCR) for *APOE* allele genotyping with AmpliTaq PCR Mix Thermo Fisher Scientific Baltics UAB V. A. Graiciuno 8, Vilnius, LT-02241 Lithuania (Thermo Cat: 4390941). Briefly, DNA sequences were amplified with *APOE* forward and reverse primers on a PCR cycling schedule of 95 °C for 10 min; 35 cycles of 95 °C for 20 s, 69 °C for 30 s, 72 °C for 45 s, 72 °C for 5 min, and 26 °C hold. The amplified product was then examined for size and quality through electrophoresis on an Agilent Tapestation D1000 Agilent Technologies Hewlett-Packard-Straße 8 76337 Waldbronn, Germany. Tapestation results were analyzed for known fragment distribution of *APOE* alleles to determine *APOE* allele status. 

### 4.5. Statistical Analyses

Statistical Package for Social Sciences version 28.0.1.1(14) (IBM SPSS Statistics for Windows, IBM Corp, Armonk, NY, USA), (https://www.ibm.com/, accessed on 1 October 2023) was used for statistical analyses. Independent two-sample *t*-tests, ANOVA, or chi-squared tests were conducted to examine group differences in age, sex distribution, IQ (KBIT-2), global cognitive function (MMSE), and self-reported autistic traits (SRS-2; Table 3). Two-way factorial, univariate general linear models were executed for each dependent variable with diagnosis group (ASD vs. NT) and *APOE* ε4 group (carrier vs. non-carrier) as independent variables and sex as a covariate. In the presence of a significant sex effect, exploratory analyses within sex and diagnosis groups were evaluated with independent two-sample *t*-tests comparing ε4 carriers vs. non-carriers. However, for autistic women, there were only two non-carriers. Therefore, a Bayesian method was conducted to compare each autistic female non-carrier to the group of autistic female carriers as a single-case comparison. SingleBayes_ES.exe was used to determine a point estimate of the percentage of the carrier population to generate a more extreme score. In addition, it evaluated the probability that a participant in the carrier population would obtain a lower score than the non-carrier [54].

## 5. Conclusions

We replicated previous findings indicating that the *APOE* ε4-allele is associated with worse verbal learning and short-term memory performance in MA+ adults. We presented preliminary results that suggest that autistic males may be particularly vulnerable to the deleterious effects of the *APOE* ε4-allele on verbal learning, but future studies with larger sample sizes (particularly of autistic women) are needed to comprehensively understand the influence of *APOE* allelic distribution on verbal learning and memory in autistic and non-autistic men and women. This is a step forward to understanding cognitive and brain aging vulnerabilities for the autistic community.

## Figures and Tables

**Figure 1 ijms-24-15988-f001:**
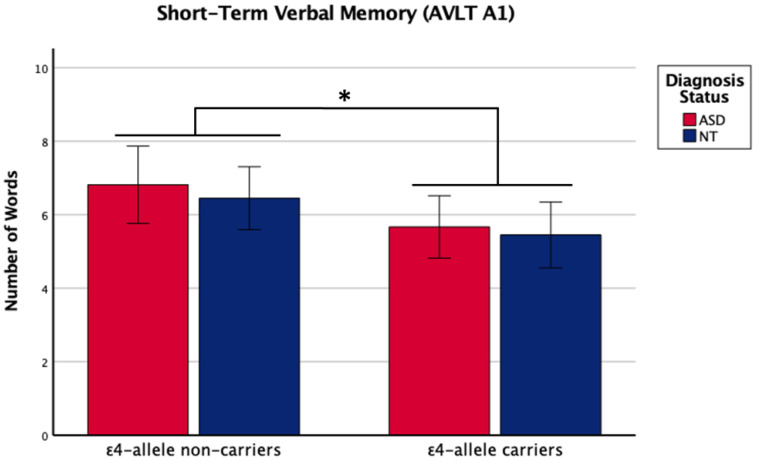
Means (± SE) by ε4-allele status and diagnosis group for short-term verbal memory on the Auditory Verbal Learning Test (AVLT A1). Sex was included as a covariate. * *p* < 0.05.

**Figure 2 ijms-24-15988-f002:**
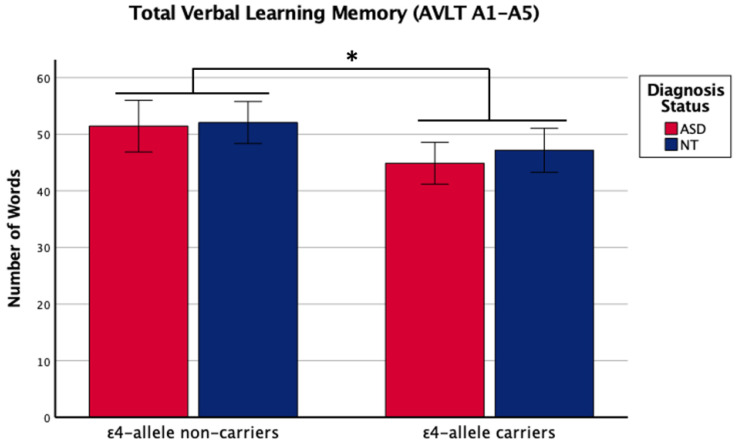
Means (± SE) by ε4-allele status and diagnosis group for total verbal learning memory on the Auditory Verbal Learning Test (AVLT A1–A5). Sex was included as a covariate. * *p* < 0.05.

**Figure 3 ijms-24-15988-f003:**
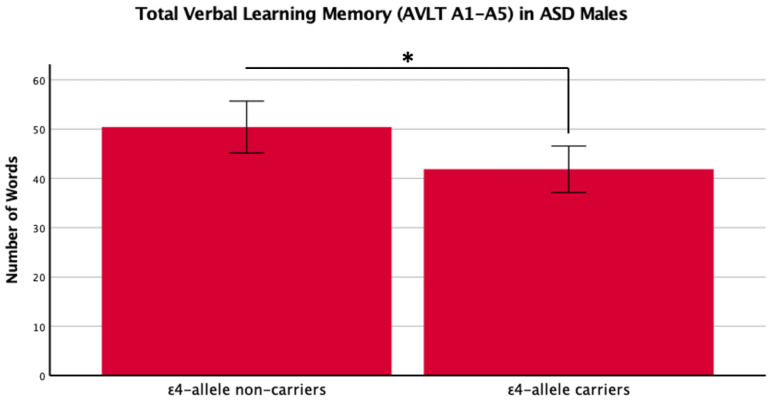
Means (± SE) by ε4-allele status in ASD males for total verbal learning memory on the Auditory Verbal Learning Test (AVLT A1–A5). * *p* < 0.05.

**Table 1 ijms-24-15988-t001:** AVLT raw trial score (A1, A1–A5, and A7) results.

	DF	*t*-Value	*p*-Value	F	Partial Eta Squared (Effect Size)
Short-Term Memory (AVLT A1)					
Diagnosis	1, 71	0.352	0.526	0.406	0.006
*APOE* ε4-allele	1, 71	1.586	**0.025 ***	5.247	0.069
Diagnosis∗*APOE* ε4-allele	1, 71	0.163	0.871	0.027	n/a
Sex	1, 71	1.37	0.177	1.863	0.026
Total Words (AVLT A1-A5)					
Diagnosis	1, 71	−0.851	0.467	0.534	0.007
*APOE* ε4-allele	1, 71	1.783	**0.006 ***	7.867	0.100
Diagnosis∗*APOE* ε4-allele	1, 71	0.420	0.676	0.176	0.002
Sex	1, 71	3.945	**<0.001 ***	15.563	0.180
Long-Term Memory (AVLT A7)					
Diagnosis	1, 71	−1.342	0.195	1.710	0.024
*APOE* ε4-allele	1, 71	0.512	0.212	1.587	0.022
Diagnosis∗*APOE* ε4-allele	1, 71	0.574	0.571	0.324	0.005
Sex	1, 71	1.710	0.86	3.038	0.041

* bold indicates *p* < 0.05.

**Table 2 ijms-24-15988-t002:** AVLT total words learned (A1–A5) within sex and diagnosis groups results.

	DF	*p*-Value	F	Partial Eta Squared (Effect Size)
ASD males	1, 25	**0.020 ***	6.183	0.198
NT males	1, 25	0.094	3.040	0.108
NT females	1, 12	0.318	1.087	0.083
Bayesian Hypothesis Single Case Comparison Test for ASD Females.				
	Case’s Test Score	Percentage of control population falling below case’s score	Effect Size	95% Confidence Interval
ASD Female Case 1	60	79.4571%	0.969	(−0.053 to 1.928)
ASD Female Case 2	37	6.6496%	−1.937	(−3.321 to −0.502)

* bold indicates *p* < 0.05.

**Table 3 ijms-24-15988-t003:** Participant demographic information and *APOE* ε4-allele carrier status.

	NT (*n* = 41) Mean (±SD) Range	ASD (*n* = 35) Mean (±SD) Range	Two-Group Comparison Statistics	NT *APOE* ε4 Carriers	NT *APOE* ε4 Non-Carriers	ASD *APOE* ε4 Carriers	ASD *APOE* ε4 Non-Carriers	Four-Group Comparison Statistics
Age (Years)	53.90 (±8.44) 40–70	53.06 (±8.91) 40–71	t(74) = 0.424, *p* = 0.673	54.05 (±7.06) 41–65	53.76 (±9.75) 40–70	54.38 (±8.50) 41–71	51.07 (±9.44) 40–67	t(75) = 0.234, *p* = 0.705
Sex (M/F)	27/14	27/8	X^2^(1.76) = 1.170, *p* = 0.279	10/10	17/4	15/6	12/2	X^2^(3.76) = 6.775, *p* = 0.079
ADOS-2 ^a^ Social Affective	n/a	10.03 (±3.12) (0–17)	n/a	n/a	n/a	10.14 (±2.78) 7–17	9.86 (±3.68) 0–14	n/a
Age at Diagnosis	n/a	46.11 (±15.35) 2–67	n/a	n/a	n/a	48.62 (±11.74) 21–64	42.36 (±19.45) 2–67	n/a
SRS-2 ^b^ Total t-score	45.39 (±5.94) 37–60	71.64 (±11.55) 43–89	t(45.435) = −11.854, *p* < 0.001	45.15 (±6.44) 37–59	45.62 (±5.57) 37–60	70.05 (±12.88) 43–89	74.08 (±9.07) 57–87	t(73) = 26.752, *p* < 0.001
MMSE ^c^	29.49 (±0.84) 26–30	29.06 (±1.11) 26–30	t(62.626) = 1.775, *p* = 0.081	29.35 (±1.04) 26–30	29.57 (±0.598) 28–30	28.90 (±1.09) 27–30	29.29 (±1.14) 26–30	t(75) = 0.849, *p* = 0.175
KBIT-2 ^d^ Composite	109.07 (±12.09) 85–141	108.97 (±14.52) 70–131	t(73) = 0.033, *p* = 0.973	106.20 (±9.38) 85–124	111.81 (±13.88) 89–141	107.14 (±13.73) 70–131	111.92 (±15.82) 83–131	t(74) = 0.487, *p* = 0.410

**^a^** Autism Diagnostic Observation Schedule-2; ^b^ Social Responsiveness Scale-2; ^c^ Mini Mental State Exam; ^d^ Kaufman Brief Intelligence Test-2.

## Data Availability

The data presented in this study are available upon request from the corresponding author. The data are not publicly available due to privacy concerns.

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
