# Peer review of "APOE ε4-Allele in Middle-Aged and Older Autistic Adults: Associations with Verbal Learning and Memory"

_ijms, 2023, doi:10.3390/ijms242115988_

Round 1

Reviewer 1 Report

Comments and Suggestions for Authors

In this manuscript, the authors present their work on determining the effect of the APOE e4 allele on verbal learning and memory of middle aged and older (MA+) autistic adults. As part of this work, the authors recruited 76 adult participants (41 neurotypical (NT) and 36 with autism spectrum disorder (ASD)). The APOE allele genotyping was done through PCR methods. The participants were administered the Rey Auditory Verbal Learning Test (AVLT) to assess short- and long-term memory and learning.

Their results show that the presence APOE e4 allele has a significant negative impact on the short-term memory and verbal learning, but had no significant effect on long-term memory. Furthermore, the results are suggestive of the fact that the APOE e4 allele has a negative effect only on males.

The major drawback of the study, as the authors themselves acknowlege, is the rather small sample size. This especially true for drawing inferences about the interaction of sex with APOE e4 allele on the verbal and learning abilities, since the study only had two female ASD participants who were negative for APOE e4 allele. However, I feel that the authors have been sufficiently cautionary about the reliability and interpretability of their results and conclusions. As an exploratory study, the data and statistical analysis are reasonable.

Clearly, more work needs to be done and a wider population must be tested to confirm the results and conclusion of this work. But beyond that, the authors should explore other demographic variations if possible. This paper is largely silent on the details of the demographics such as patient health history, ethnicity, education level, mental and physical activity levels, familial health history, etc. If any such details are available for the current study, the authors should consider reporting them though multivariate analysis of all these factors might be beyond the scope of this preliminary work.

Author Response

Kind regards,

Authors

Reviewer 2 Report

Comments and Suggestions for Authors

The paper of Dr Harker et al is a clear-cut research on verbal learning and memory done in a small sample of autistic adults. The results show that autistic men, who are carriers of the ε4-allele of the Apolipoprotein E (APOE) gene, learned worse as compared to all other groups (normotypic carriers, autistic non-carriers). The paper brings important results, which are in line with the literature.

There are a few minor point to improve the manuscript:

-        Tables’ headings are not sufficient for understanding

-        figures 1, 2 : please, indicate the sex factor in the legend

-        it would be useful to display the size of effects, since the samples are rather small

Author Response

Kind regards,

Authors

Reviewer 3 Report

Comments and Suggestions for Authors

The manuscript by Harker at al. is devoted to the study aimed to assess ε4-allele of the APOE gene on verbal functions of autistic adults. The authors hypothesized that autistic adults who carry the APOE ε4 allele would have worse results in verbal tests compared to both controls and autistic non-carriers of the APOE ε4 allele. The topic raised by the authors is certainly interesting and deserves attention. However, a number of questions need to be clarified:

The authors used unconventional approach to statistical analysis. Typically, an ANCOVA uses a continuous variable rather than a categorical variable such as gender as a covariate. It is logical to include the “gender” variable in repeated measures ANOVA as an independent factor. This is probably explained by the insufficient number of women who are non-carriers of the APOE ε4 allele (n=2).

The authors specify the use of “Two by two analysis of covariance (ANCOVA)”. Do you mean two-way factorial ANCOVA for each dependent variable? This design does not take into account whether each of the variables (for immediate and delayed word recall) belongs to the same subject. In this case, the design of repeated measures ANOVA followed by post-hoc tests is more adequate. The test type can be selected as a repeated measures factor, and carrier vs non-carrier, ASD vs NT can be selected as independent factors. If there are a sufficient number of women, gender can also be included as an independent factor.

The authors used independent two-sample t-tests or chi-squared tests to compare the control and ASD groups on a number of measurements. But are there any differences between the 4 groups that were included in the design further?

The presentation of the results is even less clear. Tables 2 and 3 provide the t-value. Are these the results of a t-test or ANCOVA?

The conclusion regarding gender differences in test performance is simply incorrect because, due to the insufficient number of female non-carriers of the APOE ε4 allele, samples of men and women were analyzed differently. Ideally, to prove this finding, the authors would need to expand the sample of women and conduct new statistical analysis.

Author Response

Kind regards,

Authors

Round 2

Reviewer 3 Report

Comments and Suggestions for Authors

Dear authors,

you did not change your statistical models, although this might have made your work more powerful regarding gender differences in the genesis of autistic disorders.

You've added careful doubt to the discussion regarding the gender of the subjects, but the abstract still contains statements about gender differences between autistic men and women. But this does not follow from your research. I recommend noting in the abstract that gender differences in men were obtained only due to undersampling of women, for whom this remains questionable.

Regarding the model for memory tests and the reluctance to use repeated measures ANOVA (although they are obtained on the same tests over time, and information about the relationship between these tests can also be a useful result), the question remains: is there significant correlation between immediate and delayed reproducing words?

Author Response

Kind regards,

Authors
